# D-Pinitol from *Ceratonia siliqua* Is an Orally Active Natural Inositol That Reduces Pancreas Insulin Secretion and Increases Circulating Ghrelin Levels in Wistar Rats

**DOI:** 10.3390/nu12072030

**Published:** 2020-07-08

**Authors:** Juan A. Navarro, Juan Decara, Dina Medina-Vera, Rubén Tovar, Juan Suarez, Javier Pavón, Antonia Serrano, Margarita Vida, Alfonso Gutierrez-Adan, Carlos Sanjuan, Elena Baixeras, Fernando Rodríguez de Fonseca

**Affiliations:** 1Laboratorio de Medicina Regenerativa, Instituto IBIMA de Málaga, Unidad de Gestión Clínica de Salud Mental, Hospital Regional Universitario de Málaga, 29010 Málaga, Spain; juan_naga@hotmail.es (J.A.N.); juandecara@uma.es (J.D.); dina.medina@ibima.eu (D.M.-V.); rubentovar7@hotmail.com (R.T.); juan.suarez@ibima.eu (J.S.); javier.pavon@ibima.eu (J.P.); antonia.serrano@ibima.eu (A.S.); mayivida@gmail.com (M.V.); 2Facultad de Medicina, Universidad de Málaga, 29010 Málaga, Spain; 3Facultad de Ciencias, Universidad de Málaga, 29010 Málaga, Spain; 4Departamento de Reproducción Animal, Instituto Nacional de Investigación y Tecnología Agraria y Alimentaria, 28040 Madrid, Spain; agutierr@inia.es; 5Euronutra S.L. Calle Johannes Kepler, 3, 29590 Málaga, Spain; 6Departamento de Bioquímica y Biología Molecular, Facultad de Medicina, Universidad de Málaga, 29010 Málaga, Spain

**Keywords:** AKT, D-Pinitol, ghrelin, insulin, insulin resistance, liver, phosphorylation

## Abstract

To characterize the metabolic actions of D-Pinitol, a dietary inositol, in male Wistar rats, we analyzed its oral pharmacokinetics and its effects on (a) the secretion of hormones regulating metabolism (insulin, glucagon, IGF-1, ghrelin, leptin and adiponectin), (b) insulin signaling in the liver and (c) the expression of glycolytic and neoglucogenesis enzymes. Oral D-Pinitol administration (100 or 500 mg/Kg) resulted in its rapid absorption and distribution to plasma and liver compartments. Its administration reduced insulinemia and HOMA-IR, while maintaining glycaemia thanks to increased glucagon activity. In the liver, D-Pinitol reduced the key glycolytic enzyme pyruvate kinase and decreased the phosphorylation of the enzymes AKT and GSK-3. These observations were associated with an increase in ghrelin concentrations, a known inhibitor of insulin secretion. The profile of D-Pinitol suggests its potential use as a pancreatic protector decreasing insulin secretion through ghrelin upregulation, while sustaining glycaemia through the liver-based mechanisms of glycolysis control.

## 1. Introduction

Insulin resistance refers to a poor response of insulin receptors through the phosphatidylinositol-3-kinase (PI3K) pathway [1]. This state demands a greater secretion of insulin by the pancreas, but without effective control on blood glucose levels. Inefficient insulin signaling can occur by several causes [1]. Usually, insulin resistance is associated with high sugar or fat diets, obesity, hyperglycemia and hyperinsulinemia. This is the prelude to type 2 diabetes mellitus (T2DM), a condition in which cells cannot respond properly to insulin [1].

Alterations in inositol metabolism are evident in situations of insulin-resistance leading to T2DM. Indeed, insulin resistance is related with changes in levels of two inositol stereoisomers, such as D-Chiro-inositol (DCI) and Myo-inositol (MI), that have been proposed to act as second messengers in insulin receptor cell signaling [2]. MI can be produced from D-glucose, and is converted to DCI by the action of a specific isomerase, so that each tissue has its own MI/DCI ratio. DCI is especially crucial for glycogen synthesis through promoting the dephosphorylation of the glycogen synthase (GS) enzyme [3,4]. In agreement with these observations, insulin resistance is associated with (i) low levels of DCI in plasma, urine, and insulin-target tissues; (ii) the excess urine excretion of MI; (iii) intracellular deficiency in MI in insulin target tissues. Moreover, administration of DCI or MI restores normal insulin sensitivity in some situations of insulin resistance [5]. However, despite all these evidences, the mechanisms through which MI and DCI exert their effects on insulin signaling are not well understood.

DCI can be synthetized endogenously in small quantities, but in humans, most DCI can be obtained from its methylated derivative D-Pinitol (3-O-methyl-chiro-inositol), which pertains to the family of inositols. It is a natural compound found in high concentrations in legumes and soy foods, and can be isolated from Bougainvillea spectabilis leaves and carob tree pods [6]. D-Pinitol exhibits important pharmacological properties, among which are its insulinomimetic effects [7,8], its beneficial effects against oxidative stress [9], and its impact on the attenuation of the effects of some pro-inflammatory cytokines [10,11]. A single dose of D-Pinitol was described to lower the plasma levels of glucose in healthy subjects, and in patients with T2DM [12,13], and long-term treatment with D-Pinitol decreased hyperglycemia and insulin levels in patients with insulin resistance [10,14,15]. Likewise, the chronic administration of D-Pinitol increased the hepatic expression of the PI3K, as well as the phosphorylation of the downstream target protein kinase B/AKT, both components of the insulin receptor-signaling pathway [16]. Moreover, treating myocytes with a PI3K inhibitor prevented the increased glucose uptake mediated by D-Pinitol [7]. Therefore, all these observations suggest that D-Pinitol participates directly in the intracellular insulin-signaling pathway. Consequently, and because it can be easily incorporated to the organism from the diet, D-Pinitol was considered a beneficial dietary supplement to prevent T2DM. However, notwithstanding all the aforementioned, other studies showed no effect of acute or chronic D-Pinitol supplementation on glycemic control [17,18,19], thus indicating that the effects and mechanisms of action of D-Pinitol on glucose metabolism remain inconclusive.

To date, most data about the effects of D-Pinitol are limited to its role in attenuating the hyperglycemia in experimental diabetic scenarios and/or in the postprandial period. No pharmacokinetics of D-Pinitol has yet been published. The present study aimed to investigate the acute effects of D-Pinitol on the insulin levels, and on several hepatic components of the insulin-mediated cell signaling in fasted Wistar rats. We focused also on hepatic glycogenesis and gluconeogenesis pathways, as well as on the secretion profile of several hormones and mediators involved in the control of the insulin-glucose metabolic homeostasis axis: leptin (whose signaling deficiency produces diabetes type 2), adiponectin (an insulin sensitizer), or ghrelin (that inhibits insulin release from pancreatic islets) [1,20,21,22].

Our hypothesis is that D-Pinitol is rapidly absorbed and distributed to the plasma and liver compartment, modulating insulin actions and contributing to glucose handling.

## 2. Materials and Methods

### 2.1. Ethics Statement

Animal experimental procedures were carried out in accordance with the European Communities directive 2010/63/EU and Spanish legislation (Real Decreto 53/2013, BOE 34/11370–11421, 2013), and approved by the Bioethics Committee for Animal Experiments of the University of Málaga, Spain, and in accordance with the ARRIVE guidelines [23]. Accordingly, all efforts were made to minimize animal suffering and to reduce the number of animals used.

### 2.2. Animals

The experiments were performed with 4- to 5-week-old male Wistar rats (Crl:WI(Han)), weighing 400 ± 20 g (Charles River Laboratories, Barcelona, Spain). The animals were kept under standard conditions (light regimen of 12/12 h, day/night) and under temperature and humidity control. The rats were fed on a standard pellet diet (STD) (3.02 Kcal/g with 30 Kcal% protein, 55 Kcal% carbohydrates and 15 Kcal% fat; purchased from Harlam (Tecklad, Madison WI)). Water and food were available ad libitum. Animals were anaesthetized with intraperitoneal (ip) sodium pentobarbital (50 mg/Kg body weight) before being sacrificed by decapitation.

### 2.3. Drug Preparation and Experimental Design

Caromax^®^-D-Pinitol (3 O methyl d chiro-inositol, 98% purity) was generously provided by Euronutra (https://www.euronutra.com/, Málaga, Spain), in the form of crystalline fine powder (lot: PPN-M0201). D-Pinitol was dissolved in water to be administered by gavage (orally) at 100 mg/Kg or 500 mg/Kg concentrations, at a volume of 1 mL/Kg. After overnight fasting, the rats were orally administered with a single corresponding dose of D-Pinitol. The rats receiving an oral dose of 100 mg/Kg D-Pinitol were sacrificed in groups (*n* = 5) at different times: 10, 20, 30, 60, 120 and 240 min after D-Pinitol load. For the oral dose of 500 mg/Kg, animals were sacrificed at times: 60, 120 and 240 min after D-Pinitol administration (*n* = 8 per group). Water was administered by gavage to control groups in a volume of 1 mL/Kg body weight. Control groups were an *n* = 5 for the 100 mg/kg dose and an *n* = 9 for the 500 mg/kg dose. They were sacrificed distributed along the different times of sampling (i.e., 0, 30, 60, 120 or 240 min), for controlling potential variations along sampling period (i.e., circadian variations). A final complementary experiment was designed to analyze the actions of D-Pinitol along a glucose tolerance test. To this end, a first set of rats were administered simultaneously with D-Pinitol 500 mg/Kg (orally) plus glucose 2 g/Kg; the second set, only had glucose 2 g/Kg. Animals were sacrificed in groups (*n* = 5–6) at different times: 30, 60 and 120 min post-administration. The glucose (Sigma-Aldrich, Saint Louis, MO, USA) was administrated intraperitoneally (i.p.), at a dose of 2 g/Kg in a volume of 5 mL per Kg of body weight in sterile saline. A control group (*n* = 6) was administered only with saline and sacrificed by being distributed along the sampling times (0, 30, 60, 120 and 240 min).

### 2.4. Sample Collection

Blood and liver samples were immediately collected. Blood was centrifuged (2100 g for 8 min, 4 °C) and the plasma was kept at −80 °C for a biochemical analysis. Liver samples were flash frozen in liquid nitrogen, then stored at −80 °C until analysis.

### 2.5. Plasma Pinitol Concentration

Plasma Pinitol concentrations were monitored by the Medina Foundation (Parque Tecnológico de Las Ciencias de la Salud, Granada 18016, Spain), using a specific liquid chromatography-mass spectrometry method. The detection of analytes and internal standards were carried out in multiple reaction monitoring mode (MRM), with electrospray positive ionization. Detection limits were 333 to 20,000 ng/mL D-Pinitol. Calculations were performed using a non-compartmental analysis of plasma data after extravascular input by, means of the computer program PK Solver 2.0 [24].

### 2.6. Measurement of Metabolites and Hepatic Enzymes in Plasma

The following plasma metabolites were measured: glucose, urea, uric acid, creatinine, bilirubin, and the hepatic enzymes glutamic oxaloacetic transaminase (GOT), glutamate pyruvate transaminase (GPT) and gamma-glutamyl transferase (GGT). These metabolites were analyzed using commercial kits, according to the manufacturer’s instructions, and a Hitachi 737 Automatic Analyser (Hitachi Ltd., Tokyo, Japan). The plasma levels of cytokines were determined by Enzyme-Linked ImmunoSorbent Assay (ELISA) method using commercial kits: leptin, adiponectin, insulin and ghrelin ELISA kits (EMD Millipore Corporation, Billerica, MA, USA, cat. number: #EZRL-83K, #EZRADP-62K, #EZRMI-13K and #EZRGRT-91K, respectively); glucagon EIA kit (Sigma-Aldrich, Saint Louis, MO, USA, cat. number: RAB0202-1KT); and an IGF 1 ELISA kit (Thermo Scientific, Waltham, MA, USA, cat. number: ERIGF1). All serum samples were assayed in duplicate within one assay, and results were expressed in terms of the particular standard hormone. The homeostasis model assessment-β (HOMA-β) was calculated following the formula HOMA-β = (20 × FINS)/(FBG − 3.5); FINS = fasting serum insulin, FGB = fasting blood glucose.

### 2.7. Glucose Tolerance Tests (GTT)

Before acute treatment, rats (*n* = 8) were food-deprived for 18 h and given a dose of 100 mg/Kg of D-Pinitol (via gavage) 240 and 30 min, before an ip injection of 2 g D-glucose/Kg. Blood samples were collected from the tail vein at 0 (basal level), 5, 10, 15, 30, 45, 60 and 120 min after D-glucose injection, and glucose concentrations were measured with a commercially available glucometer (AccuCheck, Roche, Germany).

### 2.8. Insulin Tolerance Tests (ITT)

Before acute treatment, overnight fasting rats (*n* = 8) were given a dose of 100 mg/Kg of D-Pinitol (via gavage), 1 and 2 h before an ip injection of 0.75 insulin units/Kg. Blood samples were collected from the tail vein at 0 (basal level), 5, 10, 15, 30, 45, 60 and 120 min after insulin injection, and glucose concentrations were measured with a commercially available glucometer (AccuCheck, Roche, Germany).

### 2.9. RNA Isolation and cDNA Synthesis

Total RNA was extracted from tissue sections of liver (50–80 mg) using the Trizol^®^ method, according to the manufacturer’s instructions (Invitrogen, Carlsbad, CA, USA). Total RNA was quantified using a spectrophotometer Nanodrop TM ND-1000 (Thermo Fisher Scientific, Waltham, MA, USA), to ensure A260/280 ratios of 1.8 to 2.0. Reverse transcription was carried out from 1 μg of RNA using the Transcriptor Reverse Transcriptase kit and random hexamer primers (Transcriptor RT, Roche Applied Science, Mannheim, Germany). Negative controls included reverse transcription reactions that omitted the reverse transcriptase.

### 2.10. Real-Time qPCR and Gene Expression Analysis

A real-time qPCR was performed following the criteria of the MIQE guidelines [25]. Real-time qPCR reactions were carried out in a CFX96TM Real-Time PCR Detection System (Bio-Rad, Hercules, CA, USA) as previously reported (Decara et al., 2018). The gene-specific probes for the target rat genes: *Fbp1* (Fructose 1,6 Bisphosphatase 1), *G6pc* (Glucose-6-Phosphatase Catalytic Subunit), *Pc* (Pyruvate Carboxylase), *Pck1* (Phosphoenolpyruvate Carboxykinase 1), *Pklr* (Pyruvate Kinase Liver/RBC), *Actb* (beta Actin), and *Gapdh* (Glyceraldehyde-3-Phosphate Dehydrogenase) are shown in Appendix A. All probes were obtained based on TaqMan^®^ Gene Expression Assays and the FAM™ dye label format (Life Technologies, Carlsbad, CA, USA). For all reference and target gene studies, two independent biologic samples of each experimental condition were evaluated by qPCR which, in turn, was conducted in duplicate reactions, as previously described [26]. The raw fluorescence data were submitted to the Miner algorithm available online (http://www.miner.ewindup.info/) for the calculation of respective quantification cycle (Cq) and efficiency values [27]. The target and reference gene sequence amplifications were verified to show comparable efficiencies. Repeatability between replicates was accepted when Cq values differed ≤0.7. For the relative quantification, the mean of duplicates was used. The expression of both *Actb* and *Gapdh* genes was unaffected during all experimental treatments. *Actb* gene was chosen as reference gene and the Cq values were normalized in relation to the Cq *Actb* (ΔCq). Relative quantification was calculated using the ΔΔCq method and normalized with respect to the control group. Fold gene expression values were determined using the 2 ΔΔCq method [28].

### 2.11. Protein Extraction and Western Blot Analysis

Total protein from 15–25 mg of liver samples was extracted using ice-cold cell lysis buffer for 30 min, as previously described [29]. Fifty micrograms of protein were resolved on a 4–12% (Bis-Tris) Criterion XT Precast Gels (Bio-Rad Laboratories, Inc., Hercules, CA, USA, cat. number: 3450124), and electroblotted onto nitrocellulose membranes (BioRad). For specific proteins detection, the membrane was incubated 1 h in TBS-T containing 2% BSA and the corresponding primary antibody. The phosphorylated form of proteins was determined using the corresponding rabbit anti-phospho-AKT, anti-phospho-GSK3β, anti-phospho-Glycogen Synthase, anti-phospho-mTOR, anti-phospho-IRS1 (Ser612), anti-phospho-p44/42 MAPK (ERK1/2) (Cell Signaling Technology Inc., Danvers, MA, USA) and anti-phospho-IRS1 (Tyr896) (Abcam, Cambridge, UK). The total protein was detected by using rabbit anti-AKT, anti-GSK3β, anti-Glycogen Synthase, anti-mTOR, anti-IRS1 and anti-p44/42 MAPK (Erk1/2), respectively (Cell Signaling Technology Inc., Danvers, MA, USA). Adaptin γ was detected using mouse anti-Adaptin γ (Becton, Dickinson and Company (BD), Franklin Lakes, NJ, USA). Primary antibodies were detected using anti-rabbit or an anti-mouse HRP-conjugated antibody as appropriate (Promega, Madison, WI, USA, respectively). Specific proteins were revealed using ECL™ Prime Western Blotting System (GE Healthcare, Chicago, IL, USA), in accordance with the manufacturer’s instructions. Images were visualized in ChemiDoc MP Imaging System (Bio Rad, Hercules, CA, USA). After measuring phosphorylation proteins, the specific antibodies were removed from membrane by incubation with stripping buffer (2% SDS, 62.5 mM Tris HCL pH 6.8, 0.8% ß-mercaptoethanol) 30 min at 50 °C. Membranes were extensively washed in ultrapure water, and then re-incubated with the corresponding antibody specific for the total protein. Quantification of results was performed using ImageJ software (http://imagej.nih.gov/ij). The specific signal level for total proteins was normalized to signal level of the corresponding Adaptin γ band of each sample and in the same blot. The phosphorylation stage of a protein was expressed as the ratio of the signal obtained with the phospho- specific antibody, relative to the appropriate total protein antibody. The amounts of the protein of interest in control samples were arbitrarily set as 1.

### 2.12. Cell Culture and In Vitro Experimental Design

INS-1E β-cells were cultured in complete medium containing RPMI 1640 (GIBCO) supplemented with 5% FBS (GIBCO), 1 mM Sodium Pyruvate (GIBCO), 2 mM glutamine (GIBCO), 50 µM 2-mercaptoethanol (GIBCO), 10 mM HEPES (Lonza), 100 U/mL penicillin and 100 mg/mL streptomycin (Sigma) at 37 °C, in a humidified atmosphere containing 5% CO_2_ [30]. For experiments, the cells were seeded at a density of 2.5 × 105 cells/well in 12-well plates in 2 mL of complete medium, until 80% of confluence. Then, medium was changed and cells were maintained in 1 mL of complete medium and stimulated with 3 mM or 11 mM glucose (D-(+)-Glucose Solution) for 15 h. Subsequently, cells were washed twice with glucose free complete medium and incubated for 2 h in this medium. Cell cultures were washed twice with glucose-free Krebs–Ringer bicarbonate HEPES buffer (KRBH; 135 mM NaCl, 3.6 mM KCl, 5 mM NaHCO_3_, 0.5 mM NaH_2_PO_4_, 0.5 mM NaH_2_PO_4_, 0.5 mM MgCl_2_, 1.5 mM CaCl_2_ and 10 mM HEPES and BSA 0.1 %, pH 7.4). Next, cells were incubated in 500 µL of KRBH and stimulated for 30 min with different concentrations of D-Pinitol (1 µM, 10 µM, 100 µM and 1000 µM). Control cell samples were maintained in glucose free KHRB, or in glucose 3 mM KHRB, or in glucose 11 mM KHRB. Incubation was stopped by placing the plates on ice. Supernatants were collected and their insulin content was measured by ELISA kit (EMD Millipore Corporation, Billerica, MA, USA, cat. number: #EZRMI-13K). For the extraction of whole protein lysate of the INS-1E cells, we used RIPA buffer 1X (Sigma, REF: R0278) and cOmplete™ Protease Inhibitor Cocktail (Roche, REF: 11 697 498 001). Samples were frozen-defrosted at −80 °C twice. After that, samples were centrifuged at 12.000 rpm for 15 min at 4 °C, supernatants were collected, disrupted in SDS sample buffer containing DTT and boiled for 5 min to be submitted to SDS-PAGE. Western blot technique was used for protein analysis, as previously described in the Section 2.11 of Materials and Methods.

### 2.13. Statistical Analysis

Graph-Pad Prism 6.0 software was used to analyze the data. Values are represented as mean ± standard error of the mean (SEM) of 4–10 determinations for each in vivo experimental group, according to the assay. The significance of differences within and between groups was evaluated by a one-way analysis of variance (ANOVA) followed by post-hoc test for multiple comparisons. Alternatively, for comparisons between two groups, a Student *t*-test was also used. A *p* value ≤ 0.05 was considered to be statistically significant. (* = *p* < 0.05; † = *p* < 0.01; ‡ = *p* < 0.001).

### 2.14. Data Availability

All data generated or analyzed during this study are available in a raw data file in Appendix A.

## 3. Results

### 3.1. Pharmacokinetics Analysis Shows a Rapid Absorption and Detection of D-Pinitol in Plasma and Liver of Wistar Rats after Acute Oral Load

We performed preliminary studies in plasma and liver to examine the absorption and clearance of D-Pinitol orally administrated to Wistar rats. Pharmacokinetics analysis of oral administration of D-Pinitol was obtained by monitoring plasma concentration of D-Pinitol at 0, 10, 20, 30, 60, 120, 240 and 360 min after the oral administration of a 100 mg/Kg in 18 h food-deprived male Wistar rats. As shown in Figure 1A, fasting plasma D-Pinitol concentrations were below the level of detection (minute 0). After supplementation, plasma D-Pinitol became detectable as soon as 10 min, and peaked at 60 min (Tmax) showing a rapid absorption and clearance, with a half-life (t1/2) of 100 min. Liver concentration of D-Pinitol was below detectable levels at 0, 10 and 20 min, and became detectable at 30 min after oral administration. A peak of liver concentration of D-Pinitol was observed at 120 min (Figure 1B), thus 1 h later with respect to plasma Tmax. Of note is that no accumulation was detected in liver, showing a half-life (t1/2) of 154 min, slightly greater than plasma clearance (Figure 1B), suggesting either a rapid metabolism or clearance from hepatic tissue.

When tested at a higher oral dose of D-Pinitol (500 mg/Kg), it peaked at 120 min in plasma and liver, with plasma Cmax of 54.42 ± 12.73 μg/mL, and liver Cmax of 9.87 ± 1.07 μg/mL, but still showing a rapid clearance.

### 3.2. Acute Administration of D-Pinitol Showed no Signs of Liver and Kidney Toxicities

The effects of a single dose of D-Pinitol in fasted rats were also investigated on plasma biomarkers revealing the state of the kidney or liver function [31,32]. Kidney function analysis included the study of creatinine, urea, and uric acid. After D-Pinitol (100 mg/Kg) administration at 60, 120 and 240 min, no changes in the plasma concentrations of uric acid, creatinine, and urea were detected regarding the concentrations found in the untreated control group (0 min), therefore revealing no symptoms of renal malfunction at any time after D-Pinitol administration (Appendix A). Likewise, liver function parameters included the analysis of the plasma concentration of bilirubin, and the concentration of the hepatic enzymes aspartate aminotransaminase/glutamic oxaloacetic transaminase (AST/GOT), and alanine aminotransaminase/glutamic pyruvic transaminase (ALT/GPT). After 60, 120 and 240 min of D-Pinitol (100 mg/Kg) administration, no changes in the basal levels of these hepatic enzymes and bilirubin were detected with respect to the levels found at 0 min (Appendix A).

The same set of analyses was performed in rats treated acutely with a 500 mg/Kg dose of D-Pinitol. At this dose, the plasma levels of creatinine found between 60 to 240 min after D-Pinitol load were reduced (*p* < 0.01), in comparison with levels found at 0 min. Additionally, urea levels in plasma were reduced (*p* < 0.01) after 60 min (Appendix A). Likewise, the D-Pinitol 500 mg/Kg dose reduced (*p* < 0.05) the levels of the hepatic transaminase AST/GOT at 120 min, while it increased (*p* < 0.01) the levels of ALT/GPT at 240 min (Appendix A). Despite this last observation, the global analysis of these results indicated that D-Pinitol can be considered safe for liver and kidney at a single oral administration, at the amounts of 100 and 500 mg/Kg.

### 3.3. Acute D-Pinitol Oral Administration Reduced Both Plasma Insulin Levels and Insulin Resistance Index in Fasted Rats

D-Pinitol is reported to facilitate the glucose uptake by the skeletal muscle in mouse model, acting as an insulin mimetic [33], thus its oral administration in the short-term-fasted Wistar rats was expected to reduce the glucose levels below its basal levels in plasma. Surprisingly, D-Pinitol administration did not affect glycaemia, but clearly reduced plasma insulin levels. Oral D-Pinitol intake at either 100 (Figure 2A) or 500 (Figure 2B) mg/Kg dose decreased the insulin concentration in plasma, showing levels below baseline at all times tested (*p* < 0.01 and *p* < 0.001). Unlike for insulin, the levels of glucagon were slightly increased after 60 min of 100 mg/Kg of the oral dose of D-Pinitol (Figure 2A), and significantly raised (*p* < 0.05) after 120 min of 500 mg/Kg of D-Pinitol oral load (Figure 2B). Accordingly, the analysis of the glucagon/insulin ratio in all groups tested showed a trend to increase with respect to the corresponding ratio observed at 0 min; this increase was statistically significant after 60 min (*p* < 0.05) of 100 mg/Kg D-Pinitol (Figure 2A) dose, and after 120 min (*p* < 0.05) of 500 mg/Kg dose of D-Pinitol (Figure 2B).

The hormone glucagon leads the processes of glycogenolysis and gluconeogenesis to compensate for the decrease of plasma glucose during fasting. According to the above exposed, it would be expected that the levels of circulating glucose would increase, due to the combined effect of glucagon and the decrease in insulin levels after D-Pinitol intake. However, we found that baseline glucose levels remained unchanged within the period of 0 to 240 min after the acute administration of 100 or 500 mg/Kg of D-Pinitol (Figure 2A,B). We reasoned that the decrease in insulin levels could be accompanied by an increase in insulin sensitivity by D-Pinitol treatment, therefore counteracting the effect of glucagon, and thus keeping baseline plasma glucose levels constant. Thus, the homeostatic model assessment of insulin resistance (HOMA IR) was evaluated as the insulin-sensitive range. As shown in Figure 2A,B, acute intake of D-Pinitol at doses of 100 or 500 mg/Kg, reduced significantly the HOMA IR. This effect could be observed at 60, 120 and 240 min post-oral administration of D-Pinitol. Hence, this indicated that D-Pinitol could exert a clearly beneficial effect by increasing insulin sensitivity, and probably relieving the pancreas from the burden of excessive insulin secretion.

### 3.4. D-Pinitol Reduces Insulin Secretion in INS-1 Cells

For further verification of the inhibitory action of D-Pinitol on insulin secretion, we performed a set of in vitro experiments in rat insulinoma INS 1 cells under basal conditions (3 mM glucose), or stimulated by high glucose (11 mM) glucose to insulin secretion in the absence or presence of different concentrations of D-Pinitol (1, 10, 100 and 1000 µM). The results obtained indicated that D-Pinitol partially inhibited glucose-stimulated insulin secretion in vitro (Figure 3). Therefore, these results are in agreement with the suggestion that D-Pinitol reduces insulin plasma levels in Wistar rats, by inhibiting insulin secretion in the pancreas.

The mechanism by which D-Pinitol reduces insulin secretion under high glucose conditions was further explored in vitro. High glucose increases ATP production that facilitates depolarization of insulin secretion, Ca2+ influx and activation of ERK1/2, a molecular event linked to reduced insulin production and secretion. We observed that the well described effect of high glucose (11 mM) on ERK1/2 phosphorylation was counteracted by the presence of D-Pinitol (10, 100 and 1000 µM), thus blocking a molecular event needed for insulin secretion in INS1 cells (Figure 4).

### 3.5. Acute D-Pinitol Oral Administration Induced Transient Intolerance to Glucose and Insulin

We next studied alterations in glucose tolerance in rats receiving D-Pinitol. To this purpose, the glucose tolerance test (GTT) was assessed in fasted Wistar rats orally loaded with 100 mg/Kg of D-Pinitol 30 min (DP-30 group), or 240 min (DP-240 group) before glucose (2 g/Kg) injection. A control group received only glucose injection. Values for the blood glucose at 0, 5, 10, 15, 30, 45, 60 and 120 min after glucose load were measured, and the area under the curve (AUC) was calculated for each group. As shown in Figure 5A, DP 30 group did not improve the glucose tolerance, but rather showed a delay in glucose lowering, as the blood glucose levels were statistically higher than those of the control group at 30 min after the exogenous glucose load. Indeed, the AUC in the DP-30 group was significantly higher than the AUC of the control group (Figure 5A), thus suggesting glucose intolerance in this group, although the magnitude was small (less than 20% increase). Nevertheless, it is relevant to indicate that glucose lowering efficiency in DP-30 group was similar to the control group from 45 min to 120 min, thus indicating that the glucose intolerance observed at 30 min post glucose load was transient (Figure 5A). Indeed, no major differences were observed between the DP-240 min group and the control group regarding the glucose kinetics in blood after exogenous glucose load (Figure 5A).

An insulin tolerance test (ITT) was performed to determine whether the administration of D-Pinitol (100 mg/Kg) affected the response to exogenous insulin. The ITT was assessed 1 h (DP-1h) or 2 h (DP-2hs), after the D-Pinitol administration and the blood glucose concentration were measured before and after (5, 10, 15, 30, 45, 60 and 120 min) insulin ip administration. We found that there was a diminished response (*p* < 0.01 and *p* < 0.05) to insulin in group DP-1h at 45, 60 and 120 min regarding the control group (Figure 5B). This effect was no longer observed in group DP-2hs, thus suggesting that the effect of D-Pinitol on insulin intolerance was transient. Nevertheless, no statistical difference was observed between the AUC of control group and the AUC from D-Pinitol treated groups (Figure 5B).

### 3.6. Acute Oral Administration of D-Pinitol Inhibits the Gene Expression of the Glycolytic Enzyme pyruvate Kinase in Liver

We next investigated the effect of D-Pinitol oral intake on the gene expression of key enzymes controlling glycolysis and gluconeogenesis in the liver of Wistar rats. The gene expression of the hepatic enzymes, such as pyruvate kinase (*Pklr*), glucose-6-phosphatase catalytic subunit (*G6pc*), fructose-1,6-bisphosphatase (*Fbp1*), and pyruvate-carboxykinase (*Pck1*), is normally under dietary and hormonal control. Their expression in liver was evaluated at 0, 20, 30 and 60 min after D-Pinitol load in fasted rats by PCR analysis. The results revealed that the expression of the three-gluconeogenic enzymes: *Fbp1*, *G6pc*, and *Pck1* was not altered in all groups of rats that received D-Pinitol at the times tested as compared with the control group (0 min) (Appendix A). In contrast, the analysis of the gene expression of the glycolytic enzyme *Pklr* showed a clear and sustained inhibition of transcription as soon as 10 min (*p* < 0.001) after D-Pinitol load, with a trend to recover normal levels from 60 min (*p* < 0.05) post D-Pinitol administration (Figure 6). This last observation might indicate that oral D-Pinitol treatment under fasting conditions reduced the hepatic glycolysis, probably diverting liver metabolism to an active release of glucose. This was further analyzed in the following set of experiments.

### 3.7. Acute D-Pinitol Oral Load Increases Ghrelin Levels in Plasma

The effects of D-Pinitol on peripheral hormones modulating both glucose metabolism and insulin response, (adiponectin, leptin, IGF 1 and ghrelin) were also evaluated in plasma after D-Pinitol administration. Neither adiponectin nor leptin, nor IGF 1 plasma concentrations, were affected after acute D-Pinitol load at 60, 120 and 240 min, as compared with the 0 min control group (Appendix A). In contrast, examination of the plasma levels of the hormone ghrelin, which is known to inhibit the insulin secretion in pancreatic ß-cells in vivo [34], showed that 100 mg/Kg dose of D-Pinitol was able to increase significantly the levels of ghrelin in plasma as soon as 10 min (*p* < 0.01) after D-Pinitol load, with a clear trend to maintain the levels of ghrelin above the baseline levels in plasma (Figure 7). Likewise, the higher dose of 500 mg/Kg of D-Pinitol showed a significant (*p* < 0.001) increase of this hormone in plasma 60 min and (*p* < 0.05), 240 min after D-Pinitol administration (Appendix A). Together, these results indicated that the effects of D-Pinitol on ghrelin levels are specific for this hormone.

### 3.8. Acute D-Pinitol Administration Attenuates Insulin Signaling in Liver

Next, we sought to examine the consequences of the reduced insulin circulating levels in the liver tissue of fasted rats, receiving oral administration of D-Pinitol. To this aim, we examined several key proteins of the insulin mediating signaling in the liver of rats treated with D-Pinitol. Figure 8 displays the western blot analysis of the insulin receptor substrate protein 1 (IRS1) phosphorylation state. D-Pinitol (500 mg/Kg) resulted in a clear inhibition of IRS1, since the balance between activating phosphorylation at tyrosine residues was decreased when compared with the inhibitory phosphorylation at serine residues. This is typically seen when insulin receptor is hypoactive, because of a lack of insulin activation, and it is not derived on the activity of IRS1 phosphorylation kinases modulators ERK-1 and ERK2 (Appendix A). Subsequently, we analyzed the phosphorylation state of AKT (p-AKT), showing that Pinitol (500 mg/Kg) administration reduced its phosphorylation 60 min after its administration, while the levels of total AKT protein remained unchanged (Figure 9A). The analysis also indicated that this effect was transient, since the amounts of basal p-AKT were recovered after 120 min post-D-Pinitol intake. According with a decreased activity of AKT, the phosphorylation of its substrate GSK-3ß (p-GSK3ß) was also temporarily reduced at 60 min after D-Pinitol intake (Figure 9B), thus rendering this GSK-3ß more active. When active, GSK 3ß phosphorylates and inhibits the enzyme glycogen synthase (GS). However, not only was there no increase in GS phosphorylation, but there was not even a significant reduction of GS phosphorylation observed 60 min after the administration of the inositol (Appendix A). Although not significant, the phosphorylation status of the mTOR protein, a downstream substrate of AKT, was slightly reduced at 60 min of acute D-Pinitol intake, as compared with control groups (Appendix A). All these observations in the insulin-AKT pathway were compatible with the reduced insulin secretion levels observed after D-Pinitol administration.

An additional confirmation of the role of the D-Pinitol-induced drop on insulin concentration comes from the finding that, when it is co-administered with glucose, the potent rise on insulin cannot be counteracted by this inositol, resulting in the activation of the PI3K-AKT signaling (Appendix A). A potential explanation is the well known counteractive action of glucose on ghrelin secretion.

## 4. Discussion

The search for natural ingredients that might optimize human glucose metabolism is becoming a priority, because of the current obesity pandemics. In this regard, natural inositols, as insulinomimetics, have gained attention as potential functional foods that might help to prevent the development of insulin resistance and diabetes type 2. Our current study provides new perspectives on the metabolic effects of D-Pinitol, a natural inositol from carob fruit, under fasting conditions, when circulating glucose is expected to be at baseline levels. A first innovative approach was to study the oral absorption of this natural compound, under these dietary conditions. The pharmacokinetics analysis showed that the oral D-Pinitol intake is easily absorbed and quickly detected in plasma and in liver in a few minutes, reaching a peak 60 min after its oral ingestion, and suffering a quick clearance from plasma there on, practically disappearing 6 h after its administration. Liver pharmacokinetics indicates an accumulation of this inositol in the liver, shifted in time 60 min, and with a similar rate of clearance. Thus, the half-life for plasma was 108 min and 154 min for liver. This pharmacokinetics parallels the transient effects observed on different metabolic aspects here studied and discussed below. It also accounts for the possibility of repeated administration (i.e., twice a day), in case this compound has to be used as an endocrine pancreas protector, as is deduced from its endocrinological profile, discussed below. Another important consequence of these findings is the fact that despite the potential conversion of D-Pinitol into DCI in the acid media of the stomach, a substantial amount of D-Pinitol is incorporated to the blood stream, being able to act as an active nutritional ingredient.

The analysis of plasma standard markers of renal and liver dysfunction revealed that doses of 100 and 500 mg/Kg of D-Pinitol are not toxic, being a safe compound. It should be noted that when high doses of D-Pinitol (500 mg/Kg) were tested, the levels of the transaminase GOT were transiently reduced at 120 min, while the GPT levels were increased significantly at 240 min. In this sense, repeated administration of D-Pinitol for nine weeks did not affect the transaminase profile (data not shown).

Because D-Pinitol administration seems to ameliorate the hyperglycemia in some models of diabetes, it has been considered as an insulin mimetic, like other inositols, such as myoinositol and DCI [10,12,13,14,15]. In our model in fasted rats, the main effect of D-Pinitol administration was a significant reduction of circulating insulin, resulting in an increased glucagon/insulin ratio. Since plasma glucose did not vary along the experiment, the net result is a saving of insulin, since the secretion of this hormone was reduced up to a 50% for more than 4 h. Of note is that our in vivo experiments were carried out under fasting conditions in which glucagon is supposed to play a crucial role in maintaining basal glucose blood levels through promotion of the gluconeogenesis and glycogenolysis in the liver. Therefore, after oral D-Pinitol intake, a favorable effect of glucagon on the circulating glucose levels should be expected. However, as mentioned above, D-Pinitol did not alter basal plasma glucose concentrations over 240 min in fasted Wistar rats. Other reports showing the ability of D-Pinitol in reducing plasma glucose levels in fasting conditions can be attributed to the use of different in vivo models that exhibit hyperglycemia, and with different regimens of D-Pinitol treatment. These models are usually models of animals affected by diabetes [7,14,16,35] and diabetic patients [13,15], or studies conducted after food or glucose oral uptake that demonstrate the ability of D-Pinitol in reducing the postprandial blood glucose level and stimulating GLUT4 translocation in the skeletal muscle [33]. An important limitation of the present study is the lack of information on the action of D-Pinitol in other metabolic tissues that respond to insulin, such as the adipose tissue or the brain. However, preliminary data obtained in hypothalamus indicates that D-Pinitol activates the PI3K-AKT pathway, as it does in the muscle, suggesting a tissue-specific action of D-Pinitol. However, this possibility remains to be conclusively determined.

In our model, we found that decreased insulin levels by a single D-Pinitol intake were directly related with the HOMA-IR index, thus strengthening the sensibility to insulin. In this line, studies carried out in T2DM patients that took D-Pinitol three times a day and chronically also showed reduction in the HOMA-IR index, but in these patients the D-Pinitol treatment reduced the fasting plasma glucose, while no changes in insulin or peptide C were observed [15].

Increase in glucagon activity stimulates glycogenolysis and gluconeogenesis via the cAMP/protein kinase A (PKA) activation pathway. PKA activation phosphorylates the CREB (cAMP-responsive element -binding protein) factor [36]. Among its target genes are *Fbp1*, *PCk1* and *G6pc*, all involved in gluconeogenesis. In our experiments, no changes in the transcription of these genes were observed after D-Pinitol administration, therefore suggesting that this inositol derivative did not affect the gluconeogenic via, at least at the transcription level of these genes. In contrast, rats that took D-Pinitol presented a decreased expression of the *Pklr* gene which enzyme product (Pyruvate kinase enzyme, L-PK) is closely related with the glycolysis. The L-PK is a rate-controlling glycolytic enzyme, and glucagon is known to inhibit both hepatic activity and the *Pklr* gene transcription [37,38], while insulin activates L-PK, stimulating its dephosphorylation [39,40,41]. This finding indicates that D-Pinitol leads reduction of the glycolysis pathway, probably as a consequence of the lowering insulin levels. Likewise, the attenuation in the phosphorylation of the hepatic AKT points to a lessening action of insulin in liver, which is compatible with the lower insulin levels detected in plasma.

This study also shows that acute D-Pinitol administration did not affect the levels of certain hormones controlling insulin action and energy balance like IGF-1, adiponectin, or leptin (Appendix A). Again, this is a limited set of regulatory hormones, lacking important mediators that could not be monitored, such as GLP-1, or resistin that also modulates insulin dynamics. However, interestingly, here we show, for the first time, the positive effects of single D-Pinitol intake on ghrelin levels, and this result is specific for this hormone. It is known that ghrelin inhibits the secretion of insulin in pancreatic β cells [34,42]. Further investigation is needed to decipher the mechanism of action of D-Pinitol on beta cells, to inhibit insulin secretion, as shown in our experiments with INS-1, but it is also plausible that the increase in ghrelin levels under oral D-Pinitol contributes for the transient decrease in insulin levels detected in the plasma of rats receiving a single dose of D-Pinitol. In fact, when D-Pinitol is co-administered with high glucose, that inhibits ghrelin secretion, both insulin secretion and PI3K-AKT signaling were restored, suggesting that the actions of D-Pinitol are restricted to fasting conditions.

Finally, we cannot exclude the direct actions of D-Pinitol on insulin release. In our study, the inhibitory effect of D-Pinitol on insulin secretion in the rat insulinoma INS 1 strongly suggests that D-Pinitol exerts a direct effect on the pancreatic beta cells. These in vitro observations are in agreement with the negative effect of D-Pinitol on the insulin basal levels found in fasted Wistar rats, which in turn is in line with the ITT results, showing a reduced ability of exogenous insulin to lower blood glucose in rats previously treated with D-Pinitol, even though, as we discussed above, other hormonal inputs modulating insulin secretion were also affected by this inositol. Again, the use of cell lines instead of isolated pancreatic islets is an important limitation, although INS1 rat insulinoma cells are recognized models for rat β-cells, because they retain major mechanisms of insulin secretion regulation, including the responsiveness to high versus low glucose. In fact, our data suggest the existence of a direct effect of D-Pinitol on the high glucose-induced activation of ERK1/2, a Ca2+-dependent molecular mechanism linked to the activation of insulin production and secretion [43].

In conclusion, our findings confirmed the hypothesis that D-Pinitol is an active ingredient that is rapidly incorporated into the blood stream upon its oral ingestion. The data pointed to D-Pinitol as an active nutrient, with very specific actions beyond its known incorporation to glycans acting as insulin signaling mediators. Both direct and ghrelin-mediated reduction of insulin secretion, while maintaining glycaemia, might help to protect the endocrine pancreas by alleviating pancreatic islets from the high insulin demand that occurs in insulin resistance conditions. This property makes of D-Pinitol a dietary supplement of potential utility in a pro-diabetes scenario (i.e., obesity, aging, etc.), in which the pancreas becomes exhausted due to an overproduction of an inefficient insulin. Moreover, the ability of D-Pinitol to induce a transient increase in ghrelin levels opens new perspectives on the actions and use of this inositol in pathologic situations of weight loss, such as cachexia and anorexia.

## Figures and Tables

**Figure 1 nutrients-12-02030-f001:**
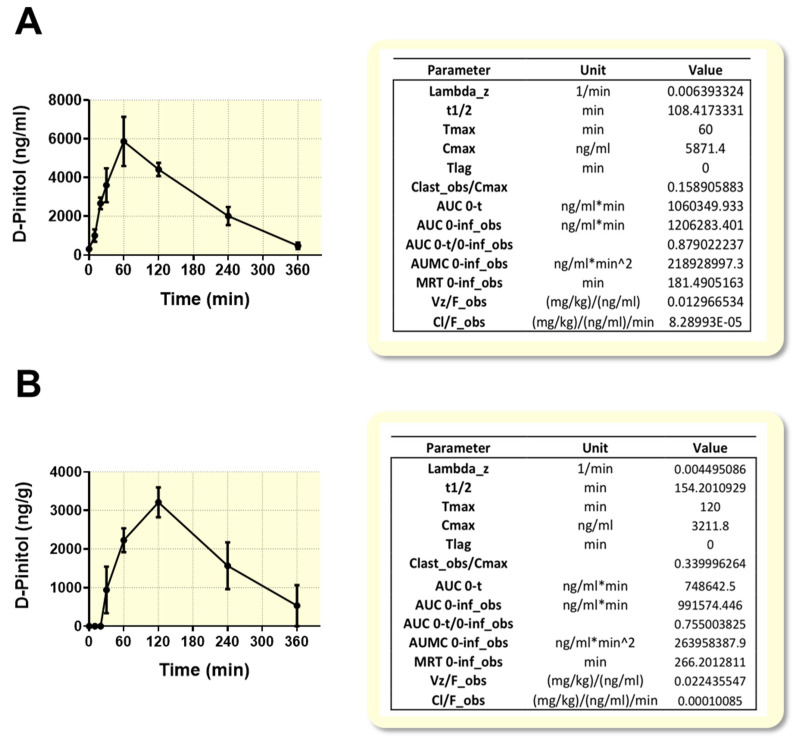
Concentration of D-Pinitol in (**A**) plasma (ng/mL) and (**B**) liver tissue (ng/g), after oral load (dose: 100 mg/kg) at different times. The values are means ± SEM (5 animals per treated group, Wistar male rats). Lambda_z: first order rate constant associated with the terminal (log-linear) portion of the curve were estimated by linear regression of time vs. log concentration. t1/2: half-life. Tmax: time of maximum observed concentration. For non-steady-state data, the entire curve is considered. For steady-state data, Tmax corresponds to points collected during a dosing interval. If the maximum observed concentration is not unique, then the first maximum is used. Cmax: maximum observed concentration, occurring at Tmax. If not unique, then the first maximum is used. Tlag: extravascular input (model 200) only. Tlag is the time prior to the first measurable (non-zero) concentration. Cl: clearance. Clast_obs: total body clearance for extravascular administration. AUC: area under the curve. AUMC 0-inf_obs: area under the first moment curve (AUMC) extrapolated to infinity, based on the last observed concentration. MRT: mean residence time. Vz: Volume of distribution.

**Figure 2 nutrients-12-02030-f002:**
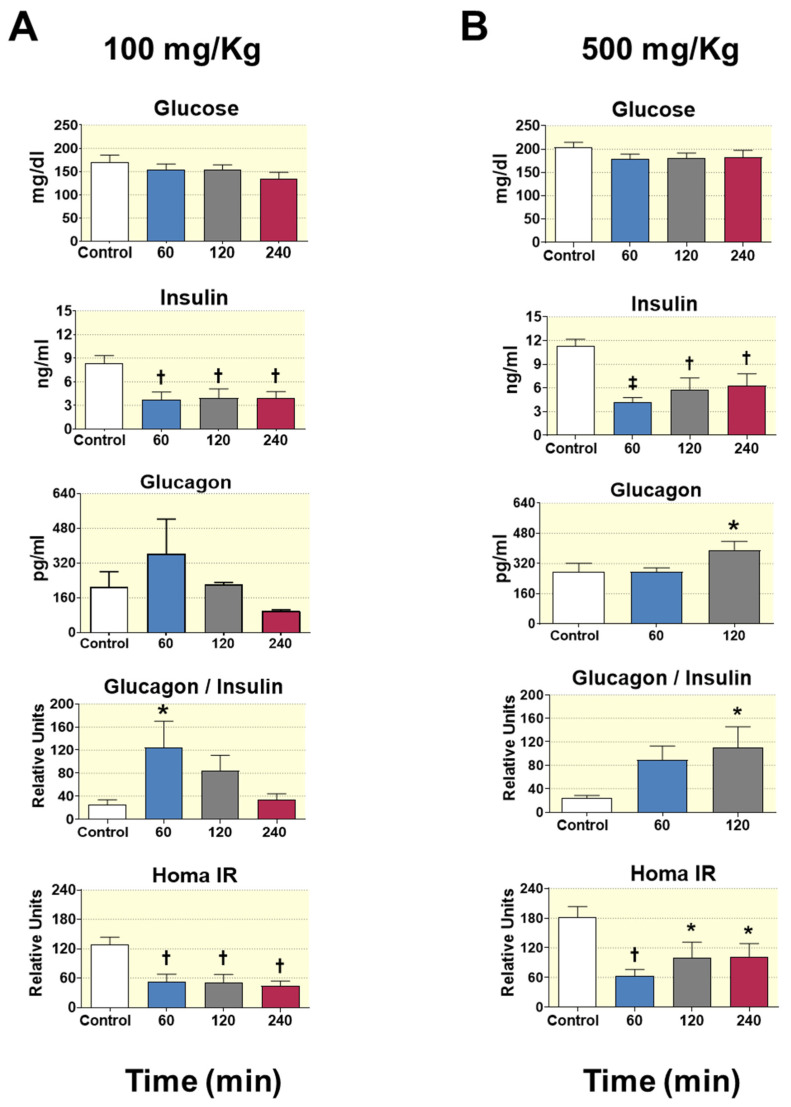
Effect of D-Pinitol on glucose, insulin and glucagon levels; on glucagon/insulin ratio and on insulin resistance index (Homa IR), at different times after administration. (**A**) Dose: 100 mg/Kg, (**B**) Dose: 500 mg/Kg. Values measured in plasma of Wistar male rats. The values are means ± SEM, 4–5 animals per group. Differences between groups were evaluated using one-way Anova + Fisher’s LSD test: * *p* < 0.05, † *p* < 0.01, ‡ *p* < 0.001 vs. Control group.

**Figure 3 nutrients-12-02030-f003:**
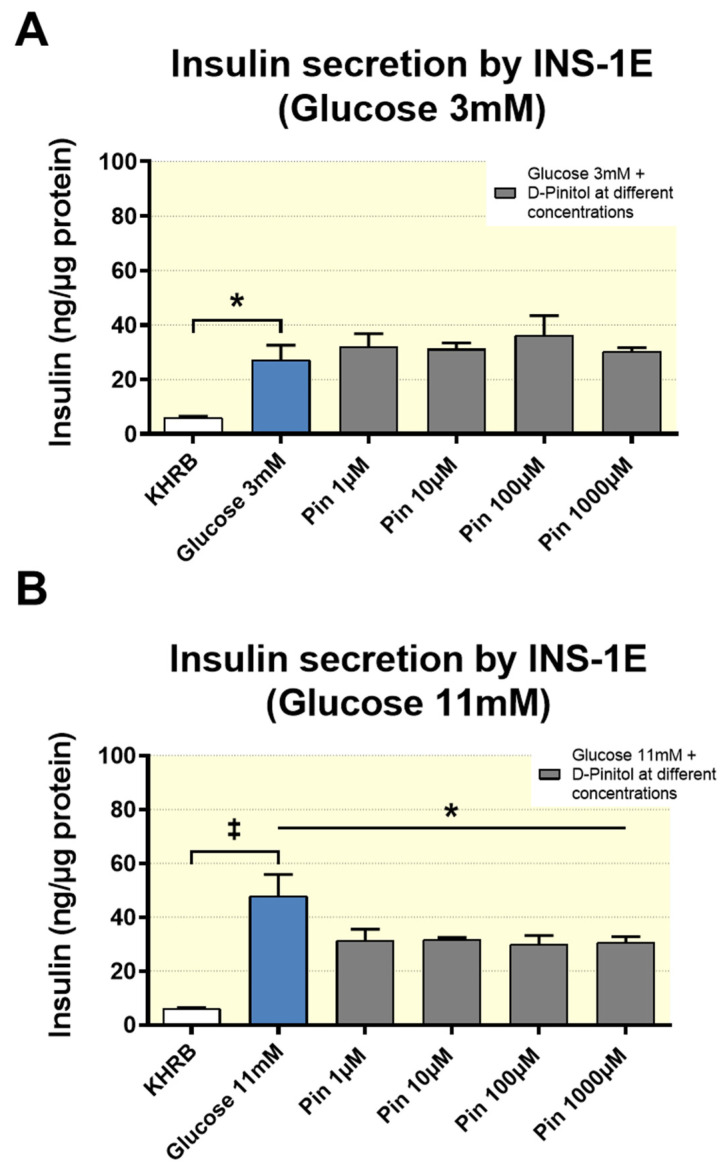
Effect of increasing concentrations of D-Pinitol (1 µM, 10 µM, 100 µM and 1000 µM) on insulin secretion by INS-1 cells under two glucose concentration conditions: (**A**) 3 mM glucose medium, (**B**) 11 mM glucose medium. Control groups shown in both conditions are represented by cell cultures treated with glucose-free Krebs–Ringer Bicarbonate HEPES medium (KRBH). The values are means ± SEM, 3 samples per treated group. Differences between groups were evaluated using one-way Anova + Fisher’s LSD test: (**A**): * *p* < 0.05 vs. KHRB medium group, (**B**): * *p* < 0.05 vs. 11 mM glucose medium group, ‡ *p* < 0.001 vs. KHRB medium group.

**Figure 4 nutrients-12-02030-f004:**
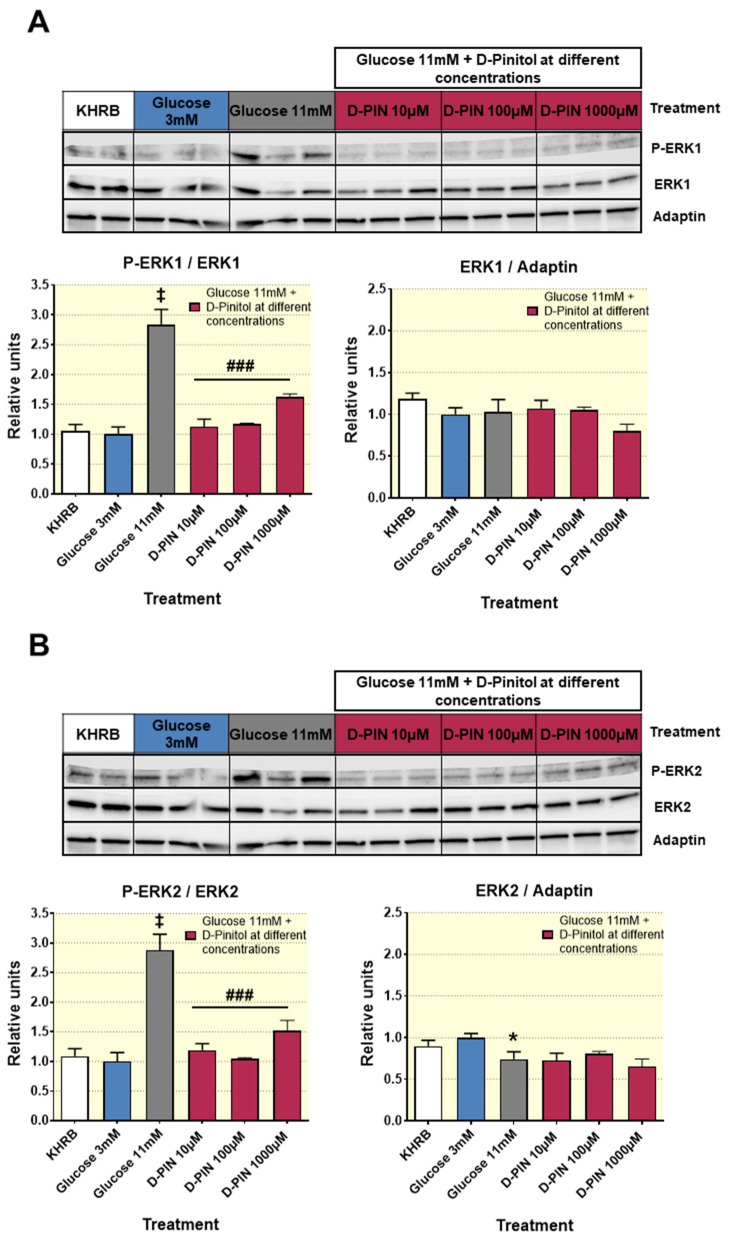
Western blot analysis of the phosphorylation status of the enzymes ERK1 and ERK2 from cell lysates of INS-1 cells treated with 3 mM glucose medium and with 11 mM glucose medium, with increasing concentrations of D-Pinitol (10 µM, 100 µM and 1000 µM). Additionally, there is a group represented by cell cultures treated only with glucose-free Krebs–Ringer Bicarbonate HEPES medium (KRBH). (**A**) Representative Western blot analysis for ERK1 band (upper panels) and p-ERK1/ERK1 ratio and ERK1/adaptin ratio (bottom panels), from cell lysates of INS-1 cells with different treatment indicated in figure. The blot shows analysis from 2–3 independent samples from each treatment group. The corresponding expression of adaptin is shown as a loading control per lane. (**B**) Representative western blot analysis for ERK2 band (upper panels) and p-ERK2/ERK2 ratio and ERK2/adaptin ratio (bottom panels), from cell lysates of INS-1 cells with different treatments indicated in figure. The blot shows analysis from 2–3 independent samples from each treatment group. The corresponding expression of adaptin is shown as a loading control per lane. All samples shown in the figure were derived at the same time and processed in parallel in the corresponding blot. The adjustment to digital images did not alter the information contained therein. Differences between groups were evaluated using one-way Anova + Fisher’s LSD test: * *p* < 0.05, ‡ *p* < 0.001 vs. Glucose 3 mM group; ### *p* < 0.001 vs. Glucose 11 mM group.

**Figure 5 nutrients-12-02030-f005:**
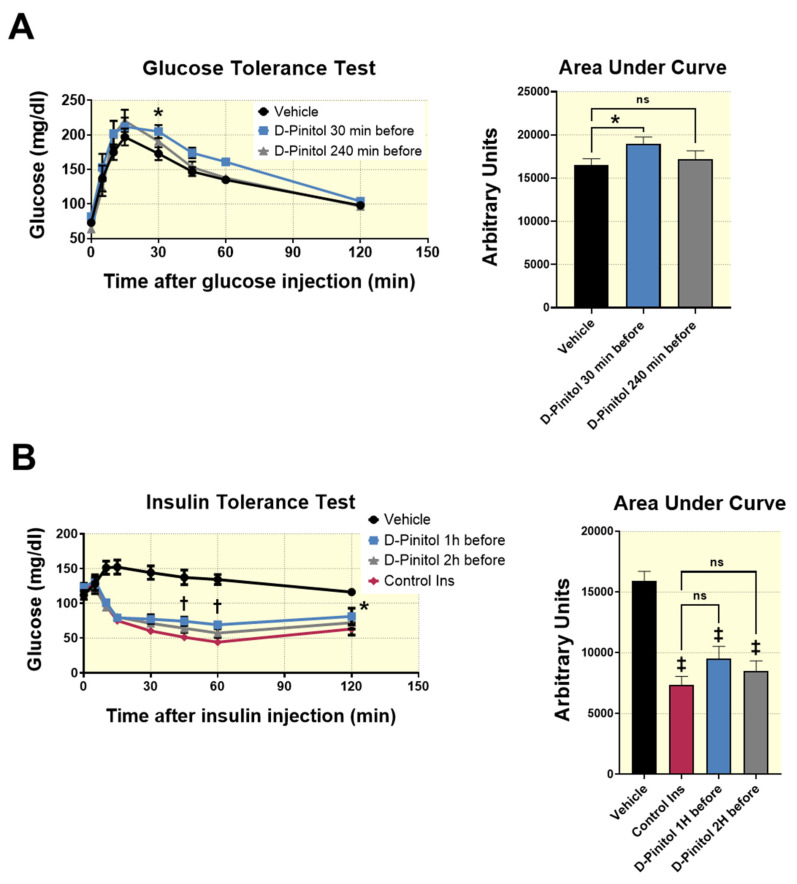
Changes in Wistar male rats blood glucose (mg/dl), during: (**A**) glucose tolerant test (GTT). Animals were fasted for 18 h before they received an i.p. injection of glucose (2 g/Kg i.p.) Blood glucose concentrations were measured in blood drawn from the tail vein using a glucometer (AccuCheck, Roche, Germany), at 0 (basal level), 5, 10, 15, 30, 45, 60 and 120 min after the glucose injection. The GTT was done at the end of 30 or 240 min of D-Pinitol (100 mg/Kg p.o.) administration. A control group (vehicle) received only glucose injection. The area under the curve (AUC) was calculated for each group. The values are means ± SEM, eight animals per group. Data were analyzed using two-way Anova + Fisher’s LSD test: * *p* < 0.05 D-Pinitol 30 min group vs. vehicle group, ns = not significant. (**B**) Insulin tolerance test (ITT). Animals were fasted for 18 h before they received an i.p. insulin injection (0.75 insulin units/Kg). Blood glucose concentrations were measured in blood drawn from the tail vein using a glucometer (AccuCheck, Roche, Germany) at 0 (basal level), 5, 10, 15, 30, 45, 60 and 120 min after the insulin injection. The ITT was done at the end of 1 or 2 h of D-Pinitol (100 mg/Kg p.o.) administration. Vehicle group received only a saline solution injection; Control Ins group received only insulin injection. The area under the curve (AUC) was calculated for each group. The values are means ± SEM, eight animals per group. Data were analyzed using two-way Anova + Fisher’s LSD test: * *p* < 0.05 D Pinitol 1h group vs. Control Ins group, † *p* < 0.01 D-Pinitol 1h group vs. Control Ins group, ‡ *p* < 0.001 vs. Vehicle group, ns = not significant.

**Figure 6 nutrients-12-02030-f006:**
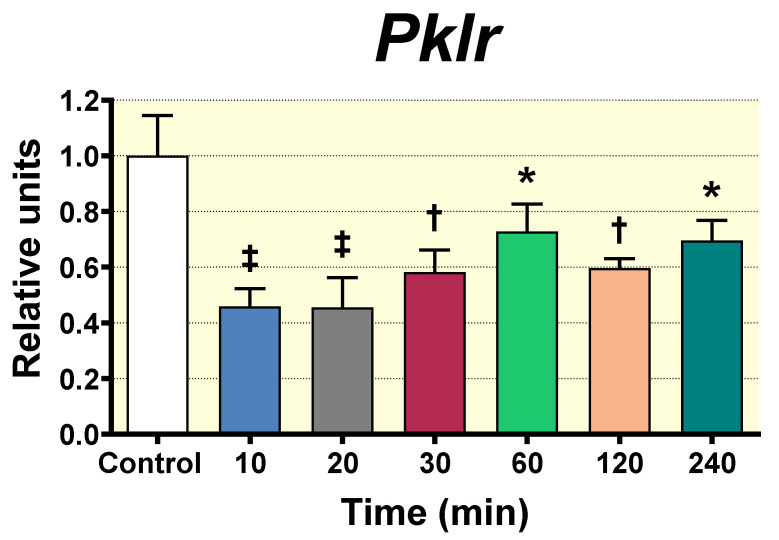
qPCR analysis of *Pklr* gene expression in liver tissue of Wistar male rats measured at different times (10, 20, 30, 60, 120 and 240 min) after D-Pinitol treatment (100 mg/Kg p.o.). The values are means ± SEM, 4–5 animals per group. Differences between groups were evaluated using one-way Anova + Fisher’s LSD test: * *p* < 0.05, † *p* < 0.01, ‡ *p* < 0.001 vs. Control group.

**Figure 7 nutrients-12-02030-f007:**
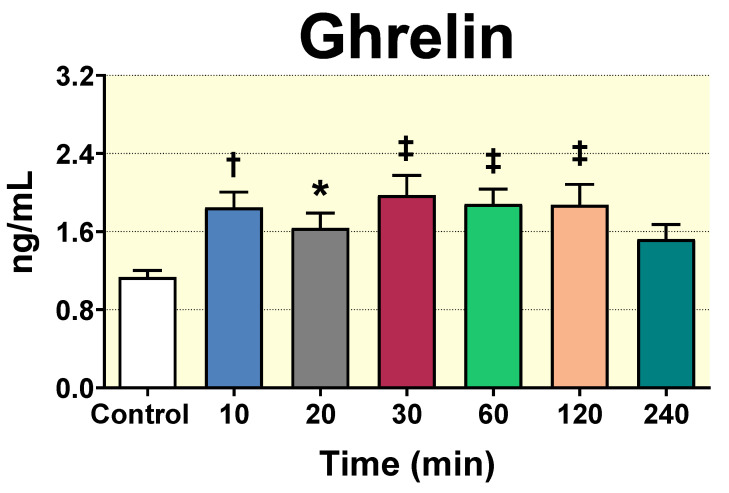
Effect of D-Pinitol (100 mg/Kg p.o.) on ghrelin levels (ng/mL) in plasma of Wistar male rats at 10, 20, 30, 60, 120 and 240 min after administration. Values measured using a commercial ELISA kit. The values are means ± SEM, 3–5 animals per group. Differences between groups were evaluated using one-way Anova + Fisher’s LSD test: * *p* < 0.05, † *p* < 0.01, ‡ *p* < 0.001 vs. Control group.

**Figure 8 nutrients-12-02030-f008:**
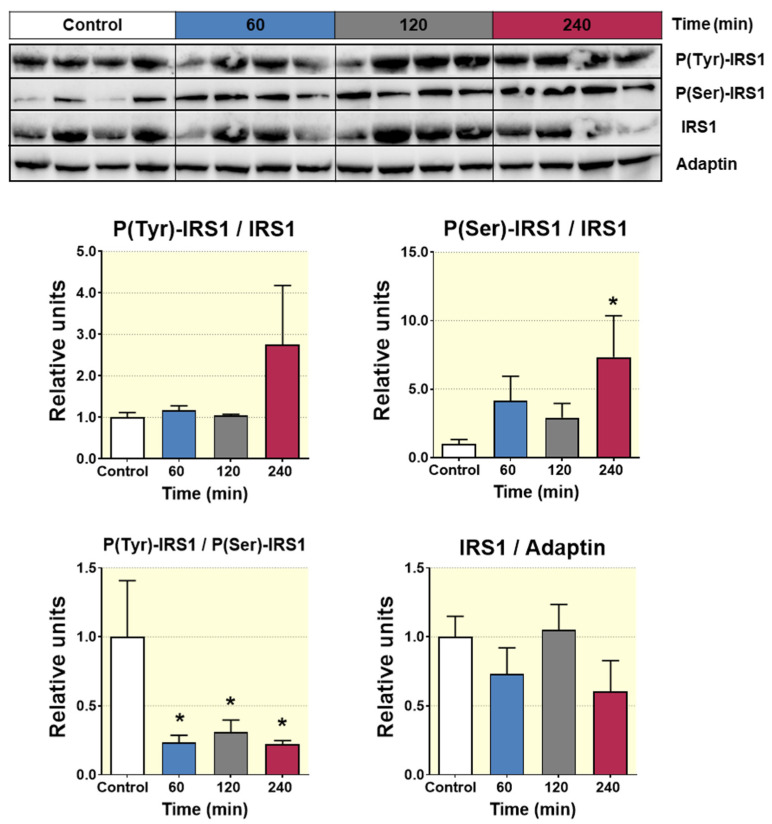
Western blot analysis of the phosphorylation status of the IRS1 from liver lysates of Wistar rats treated with 500 mg/Kg of D-Pinitol (p.o.) for 60, 120 and 240 min. Representative Western blot analysis for IRS1 (upper panels) and p(Tyr)-IRS1/IRS1 ratio, p(Ser)-IRS1/IRS1 ratio, p(Tyr)-IRS1/p(Ser)-IRS1 ratio and IRS1/adaptin ratio (bottom panels) from liver samples of Wistar rats treated with D-Pinitol for times indicated in figure. The blot shows analysis from four independent samples from each treatment group. The corresponding expression of adaptin is shown as a loading control per lane. All samples shown in the figure were derived at the same time, and processed in parallel in the corresponding blot. The adjustment to digital images did not alter the information contained therein. Differences between groups were evaluated using one-way Anova + Fisher’s LSD test: * *p* < 0.05 vs. Control group.

**Figure 9 nutrients-12-02030-f009:**
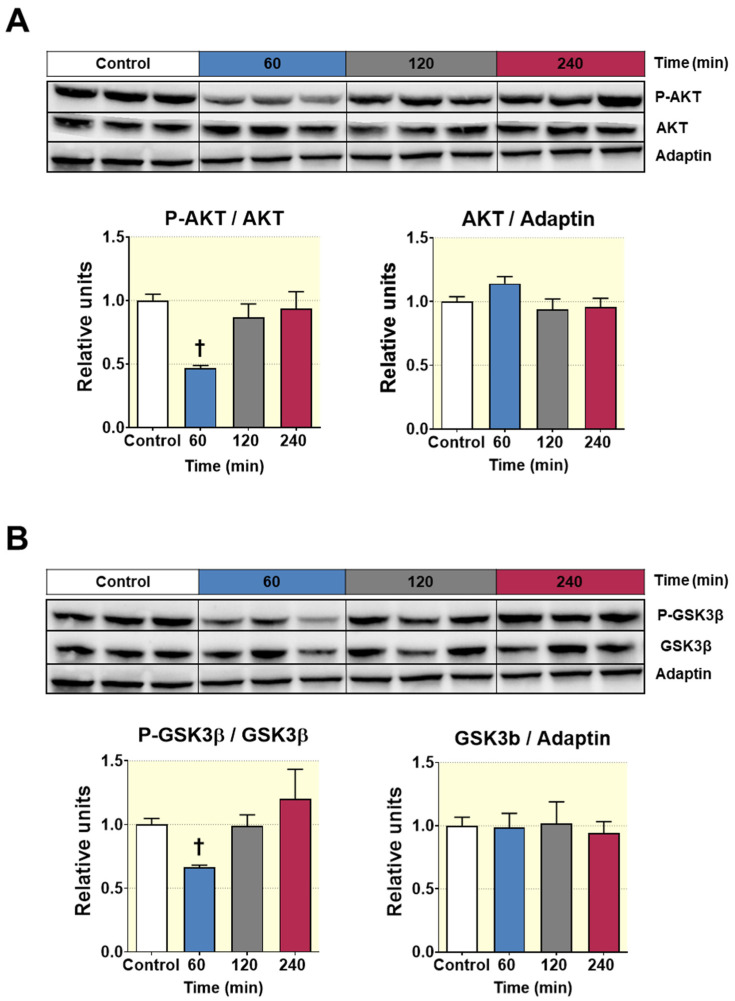
Western blot analysis of the phosphorylation status of the enzymes AKT and GSK3β from liver lysates of Wistar rats treated with 500 mg/Kg of D-Pinitol (p.o.), for 60, 120 and 240 min. (**A**) Representative western blot analysis for AKT (upper panels) and p-AKT/AKT ratio and AKT/adaptin ratio (bottom panels), from liver samples of Wistar rats treated with D-Pinitol for times indicated in figure. The blot shows analysis from three independent samples from each treatment group. The corresponding expression of adaptin is shown as a loading control per lane. (**B**) Representative western blot analysis for GSK3β (upper panels) and p-GSK3β/GSK3β ratio and GSK3β/adaptin ratio (bottom panels) of liver samples from Wistar rats treated with D-Pinitol, at times indicated in figure. The blot shows analysis from three independent samples from each treatment group. The corresponding expression of adaptin is shown as a loading control per lane. All samples shown in the figure were derived at the same time and processed in parallel in the corresponding blot. The adjustment to digital images did not alter the information contained therein. Differences between groups were evaluated using one-way Anova + Fisher’s LSD test: † *p* < 0.01 vs. Control group.

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
