# Peer review of "D-Pinitol from Ceratonia siliqua Is an Orally Active Natural Inositol That Reduces Pancreas Insulin Secretion and Increases Circulating Ghrelin Levels in Wistar Rats"

_nutrients, 2020, doi:10.3390/nu12072030_

Round 1

Reviewer 1 Report

It is difficult to understand the effects of Pinitol administration, as authors present insufficient data to clarify its effects on insulin action. In Figure 2, for example, Pinorol decreased insulin levels and reduced HOMA-IR, showing increased insulin sensitivity. In contrast, there was a diminished response to insulin in group after D-Pinitol administration during ITT study in Figure 4, and reduced insulin signaling, the phosphorylation of AKT and GSK-3ß in Figure 7. Assessment of the direct in vitro effects of Pinitol on insulin target organs such as liver, skeletal muscle and adipose tissue would support it. Furthermore, although authors suggests that the attenuation in these phosphorylation is due to a lessening action of insulin caused by the lower plasma insulin levels, the effect of Pinitol on the phosphorylation of hepatic insulin signaling molecules should be assessed upon administration of exogenous insulin (ITT).

Regarding the insulin secretion, it is necessary to examine blood insulin secretion in GTT. For completeness, it is better to examine the effect of Pinorol on the physiological insulin secretion using rat isolated islets rather than INS-1E cell line, and further discussed the mechanistic insight in beta cell.

Reviewer 2 Report

The study by Juan A. Navarro et al entitled “D-Pinitol from Ceratonia siliqua is an orally active natural inositol that reduces pancreas insulin secretion and increases circulating ghrelin levels in Wistar rats” has been reviewed. The manuscript aims to characterize the metabolic actions of D-Pinitol, a dietary inositol, in male Wistar rats.

In general the manuscript contains a lot of valuable information, and it is properly written.

  1. My first concern is the lack of justification to asses ghrelin, leptin and adiponectin, why those markers, instead of any other? Please provide the rationality of such decision in the introduction.
  2. Why do the authors repeated the measurements 7 times after the 100 mg/Kg dose and why only 2 times for the 500 mg/kg dose?
  3. The authors did not describe in methods the measurements performed 1440 minutes after ministrations. Please add the description in methods.
  4. The authors stated in methods that Water was administered by gavage to 107  control group (n=8). Due the 2 administrated doses, measured at 7 and 2 time points. How many rats were used as controls?
  5. In general, all the graphics are missing the controls response. Please add it.
  6. Figure 2. Please merge graphs of 100 and 500 mg/kg in the same plot to have a properly comparison, or at least use the same y-axes scale.
  7. Figure 2. Control is missing. Please add.
  8. Figure 5. Control is missing. Please add.
  9. Figure 6. Control is missing. Please add.
  10. Authors states that D-pinol generates a transient effect on glucose metabolism; nevertheless, according to figure 6, 1440 minutes after the administration, the effect remains, that does no seems a transient effect; actually it looks almost exactly the same effect as 10 minutes after administration!. Please explain.
  11. Another mayor concern is the use of HOMA-B, in a murine model. Please justify.
  12. Please add a supplementary table with the raw data.

Round 2

Reviewer 1 Report

Regarding the inhibitory effect of pindolol on insulin secretion, it seems that 11mM glucose-induced Ca elevation is enhanced by Pindolol in mouse islets as shown in the attached figures. Although authors are going to try the signalling in beta cells as independent approach in future study, they should show it at least by Ca measurement in INS-1 cells.

Reviewer 2 Report

This reviewer has no further comments.

Author Response

We have provided new data on the mechanistics by which D-Pinitol reduces secretion on insulinoma cells: Blockade/attenuation of high glucose-induced ERK1/2 phsophorylation, an event linked to reduced insulin secretion and insulin production.